

# A pan-European analysis of large-scale drivers of severe convective outbreaks

Monika Feldmann[1], Daniela I.V. Domeisen[2,3], and Olivia Martius[1]

[1]Institute of Geography - Oeschger Centre for Climate Change Research, University of Bern, Switzerland
[2]Institute of Earth Surface Dynamics, University of Lausanne, Lausanne, Switzerland
[3]Institute of Atmospheric and Climate Science, ETH Zürich, Zurich, Switzerland

**Correspondence:** Monika Feldmann (monika.feldmann@unibe.ch)

**Abstract.** Severe convective outbreaks have been an important driver of weather-related damages in Europe in recent years. Regional convection affecting thousands of square kilometers is driven by large-scale conditions that establish convectively favourable conditions. Systematically analysing the large-scale drivers of severe convective outbreaks helps link synoptic-scale predictability to convective-scale hazards, addressing persistent challenges in forecasting and impact assessment. We analyse the continental-scale atmospheric and land-surface conditions in the days leading up to widespread severe convective outbreaks in Europe with reanalysis data. We split Europe into regions that often experience severe convective outbreaks on the same day. Each region shows distinct dynamical and thermodynamic patterns leading up to an outbreak. Colder regions north of the Alps are associated with synoptic-scale upper-level wave patterns, accompanied by strong temperature anomalies, as they can be considered to be temperature-limited. Severe convection in drier regions of eastern Europe is associated with greater moisture anomalies. Severe convection in regions bordering the Mediterranean are associated with weak upper-level flow anomalies. These regions have a climate climate that is favourable for convection and convection is more frequent. The required additional contribution from the upper-level is thus weaker.

## 1 Introduction

Severe convective storms are one of the most expensive weather hazards in recent years, even the most expensive in Europe (Bowen et al., 2024), and have been increasing in frequency and severity over time (Rädler et al., 2019) with related costs steadily increasing (Hoeppe, 2015). Especially widespread outbreaks that cover thousands of square kilometres are responsible for severe weather hazards such as widespread hail, flooding, and wind damage, causing serious societal impacts (Bowen et al., 2024). In this study, we analyse the synoptic conditions associated with widespread severe convective environments in different regions of Europe. Understanding the large-scale drivers is a key step toward learning the typical formation mechanisms of severe convective outbreaks and ultimately improving their predictability.

While previous studies have analysed the synoptic conditions tied to convection in single European regions (Wapler and James, 2014; Morris, 1986; Mohr et al., 2019; Piper et al., 2019; Feldmann et al., 2021; Barras et al., 2021; Nisi et al., 2018), there exists no systematic documentation of synoptic conditions across all of Europe, especially derived from a homogeneous dataset. Many studies use pre-classified weather regimes (Wapler and James, 2014; Feldmann et al., 2021; Barras et al., 2021; Nisi et al.,



2018; Augenstein, 2025), which are very specific to the targeted region.

For Western Europe, the Spanish plume is an established synoptic driver of severe convection (Morris, 1986). This pattern includes the approach of a cyclone, as well as the presence of an elevated dry layer stemming from either the Iberian peninsula or North Africa. This dry layer increases convective inhibition, preventing an early onset of convection during the day. This evolution allows the convective environment to become more severe through surface heating before convection ultimately sets

in. For central Europe, southwesterly mid-level winds are commonly associated with severe convection (Wapler and James, 2014; Mohr et al., 2019; Feldmann et al., 2021). However, hailstorms and, in particular, multi-day hail clusters are associated with westerly winds (Nisi et al., 2018; Barras et al., 2021). In the Mediterranean, short-wave cutoffs are known to cause extreme precipitation events that are tied to stationary thunderstorms (Portal et al., 2024; Faranda et al., 2024). Mediterranean cyclones play a considerable role in the occurrence of convection, especially in autumn. Northern Italy, the most active convective region

in Europe, has a climate favourable to convection (Taszarek et al., 2019; Kahraman et al., 2024; Cui et al., 2024). Severe hail is often associated with low pressure systems North of the Alps (De Martin et al., 2025).

A typical model of severe convection, especially tornadoes, has been established for the USA, where the prefrontal zone of a cold front is particularly beneficial to severe convection and tornado formation, due to the enhancement of deep layer and low level shear, owed to the positioning of the low level, subtropical and polar jet streams, as well as moisture and warm air

advection in the warm sector (Doswell, 2001; Doswell and Bosart, 2001; Barnes, 1978; Barnes and Newton, 1986). While the proposed framework is not necessarily specific to the USA, we lack quantitative analyses for Europe, discussing the regional applicability of this framework.

With these regional assessments in mind, we present a comprehensive analysis of conditions associated with widespread convection in different regions of Europe based on 43 years of ERA-5 reanalysis data from May to September (Hersbach

et al., 2020). We focus on the extended summer months as the primary convective season in Europe (Taszarek et al., 2019; Giordani et al., 2024, autumn convection in the Mediterranean is mostly over the ocean). We split Europe into different sub-regions that experience convective conditions on the same day and compute composites of the atmospheric state on the first day of an event. Further analyses target differences between transient and persistent events. Finally, we inspect the influence of underlying trends.

## 2   Data and Methods

We use ERA-5 reanalysis data (Hersbach et al., 2020) from 1980-2023, aggregated daily and focusing on the extended summer months, May - September. The variables, their resolution and aggregation method are listed in Table 1. As ERA-5 parametrises convection and does not assimilate precipitation, convection itself is not well represented (Hersbach et al., 2020). We therefore define a convective proxy using maximum convective available potential energy (CAPE) and average 900 - 500 hPa bulk wind

shear. We define a convective index (CIX) at 500 J kg$^{-1}$ CAPE and 10 m s$^{-1}$ bulk shear on the same day to indicate organised convection of at least multicellular mode (see Eq. 1; Taszarek et al., 2017; Brooks, 2009).





$$CIX = \begin{cases} 1, & \text{if CAPE} > 500 \text{ J kg}^{-1} \text{ and bulk shear} > 10 \text{ m s}^{-1} \\ 0, & \text{otherwise} \end{cases} \tag{1}$$

**Table 1.** Overview of variables

| Variable | abbreviation | units | daily aggregation | resolution |
|---|---|---|---|---|
| Convective available potential energy | CAPE | J kg$^{-1}$ | maximum | 0.25° |
| 2m temperature | t2m | °C | average | 0.25° |
| convective precipitation | cp | mm | sum | 0.25° |
| 0-7 cm soil moisture | swvl1 | l m$^{-3}$ | average | 0.25° |
| 2m relative humidity | r2m | % | average | 0.25° |
| 500 hPa geopotential height | z500 | m | average | 0.5° |
| 500 hPa horizontal wind | v500 | m s$^{-1}$ | average | 0.5° |
| 900 hPa relative humidity | r900 | % | average | 0.5° |
| 900 hPa specific humidity | q900 | g kg$^{-1}$ | average | 0.5° |
| 900 hPa horizontal wind | v900 | m s$^{-1}$ | average | 0.5° |
| bulk shear | bs | m s$^{-1}$ | average | 0.5° |

## 2.1 Convective Regions in Europe

To obtain a regional division for convective events, we use K-means clustering (Hartigan and Wong, 1979). This approach
differs from typical weather regimes, which are obtained by applying an aggregation method to the synoptic pressure field,
specific to a particular region (Weusthoff, 2011; Hochman et al., 2021, here for Switzerland and Western Europe). We strive to
obtain cohesive regions throughout Europe that experience convection on the same day, to subsequently identify their primary
large-scale drivers associated with severe convection. By focusing on different convective regions, we can focus the analysis of
the synoptic scale on only convectively relevant events, rather than all occurring weather situations. We use the binary CIX in
K-means clustering (Hartigan and Wong, 1979) to identify regions that experience considerable CAPE and shear on the same
day. Sensitivity experiments with a range of different numbers of clusters showed the meteorologically most cohesive regions at
10 clusters (see Section 3.1 for discussion). One out of these 10 clusters depicts convectively relatively inactive areas, and one
cluster covers southeastern Spain and Northern Africa. As we focus on Europe in this study, this cluster is not analysed further.
The remaining 8 clusters that we will focus on here are shown in Fig. 1a. There are four regions bordering the Mediterranean
and four regions located north of the Alps. We also experimented with rectangular regions based on lightning density data.
These regions yielded similar results for synoptic atmospheric and land-surface anomalies, emphasising the robustness. We
nonetheless preferred to use a meteorologically informed, objective method of region identification over rectangular boxes.





## 2.2 Convective Event Analysis

Within these regions, we define severe convective outbreaks (SCO) as a minimum area of 100 grid points with a positive
CIX, indicating widespread convection. Additionally, convective precipitation must be non-zero somewhere in the CIX area
to ensure the presence of a convective trigger. Due to uncertainties regarding the convective parametrization (Hersbach et al.,
2020), we do not use convective precipitation as a direct proxy of convection; convective precipitation is only an indication of
triggering (Taszarek et al., 2020). The results are not particularly sensitive to the chosen thresholds. We experimented with a
range of CAPE (300-1000 J kg$^{-1}$) and bulk shear values (10-20 m s$^{-1}$), all yielded qualitatively the same conclusions. The
80 values of the thresholds modulate the number of detected events, as well as the absolute magnitude of the anomalies.

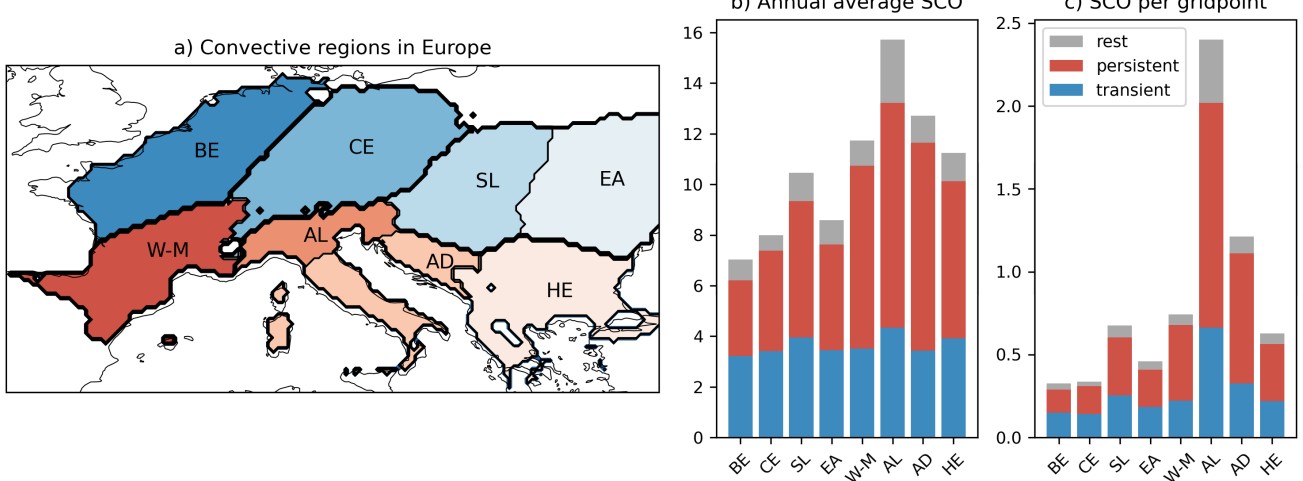

**Figure 1.** Identified convective regions with K-means clustering and their convective activity. Panel b) shows the total number of SCO per regional grid point, panel c) shows the number of annual SCO per aggregated region.

The SCO classification provides a binary daily event time series per region, showing all days that meet the SCO definition. We trim this down to only contain the first days for those episodes that last multiple days. We additionally split the event dataset into single-day outbreaks and multi-day outbreaks to analyse the difference between persistent SCO and transient SCO. Transient is defined as a single SCO day followed by at least two non-SCO days. Persistence requires at least two consecutive
SCO days. Events that meet neither criterion are classified as "rest". An overview of all identified SCO is provided in Fig. 1. All descriptor variables are deseasonalised using a 10-day rolling window. For each region, we then create time-lagged composites that show the mean atmospheric conditions leading up to the first day of all SCO for each region (see Section 3.2). We also derive the composites for transient and persistent SCO separately in the same manner and depict them as Hovmöller diagrams, averaging across the latitude band of each region (see Section 3.3).
Statistical testing of anomalies is performed with a Mann-Whitney-U test, comparing all non-SCO days to the first SCO-day.



Subsequently, false-discovery-rate (FDR) correction is applied to reduce artefacts introduced by repeating testing for every grid point (Ventura et al., 2004; Benjamini and Hochberg, 1995). For the Hovmöller diagrams, the aggregation of p-values is performed with the Stouffer Z-score method, which accounts for spatial dependencies, after the FDR correction (Stouffer et al., 1949).

Lastly, climatological trend analyses are performed using a linear trend fit on annual averages (see Section 3.4). Significance testing of the linear trends is done with the Hamed-Rao modification of the Mann-Kendall trend test, which accounts for annual lag-1 temporal autocorrelation (Hamed and Ramachandra Rao, 1998; Hussain and Mahmud, 2019).

## 3 Results

### 3.1 Convective regions of Europe and their summer climate

Based on K-means clustering, we identify 8 distinct convective regions in Europe, as shown in Fig. 1a. They can be split into regions adjoining the Mediterranean, from the Western Mediterranean (W-M), over Alta-Italia (AL) and Adria (AD) to the Hellenic Peninsula (HE), and regions north of the Alpine divide, with the Benelux (BE) area, Central Europe (CE), the Slavic area (SL), and Eastern Europe (EA). This split aligns well with observed convective outbreaks, e.g., thunderstorms initiating in western Switzerland, moving into Germany and experiencing upscale growth and finally decaying in Poland (e.g., Mohr
et al., 2020; Wilhelm et al., 2021; Kopp et al., 2023), mesoscale convective systems in southern France (Arnould et al., 2025), derechos in northern France and Belgium (Fery and Faranda, 2024), or record-breaking hailstorms in northern Italy (Eisenbach, 2023; De Martin et al., 2025; Manzato et al., 2022; Bagaglini et al., 2021).

The Mediterranean regions experience a higher frequency of SCO, as well as a higher fraction of multi-day SCO (Fig. 1b). Particularly Alta-Italia is notable, as it not only has the highest number of SCO but also the smallest area, resulting in an
110 exceptionally high grid-point-normalised SCO frequency (Fig. 1c). It is almost twice as high as in the neighbouring Adriatic. Contrastingly, the midlatitude regions experience much lower absolute and area-normalised frequencies of SCO. Benelux and Central Europe show the lowest SCO frequency. To better understand the differing magnitude of large-scale descriptors for SCO throughout Europe, we inspect the climatology of the CIX, CAPE, temperature, and relative humidity from May-September, 1980-2022 (see Fig. 2). Here, we focus on the average instability, as well as its prime ingredients, surface temperature, and
115 moisture. Provided that sufficient moisture is available, CAPE is highly correlated with surface temperature (Emanuel, 2023).

There is a clear meridional gradient for the CIX, CAPE, and 2m temperature, increasing southwards, whereas 2m relative humidity is more heterogeneous, reflecting the topography (see Fig. 2). Western and Central Europe have a climate that is considerably cooler than the Mediterranean but also than eastern continental Europe in the summertime. Instability is related to temperature, so the likelihood of widespread instability when absolute temperatures are high (Emanuel, 2023). Consequently,
average CAPE values are smallest in Western and Central Europe and highest in Alta-Italia. The low instability indicates that Western Europe needs to depart far from its average conditions to support widespread convection. Surrounding the Mediterranean, unstable conditions are generally much more frequent, and the diurnal cycle alone can be a sufficient trigger for convection. The Slavic and Eastern European areas are exposed to a more continental climate, with higher temperatures but





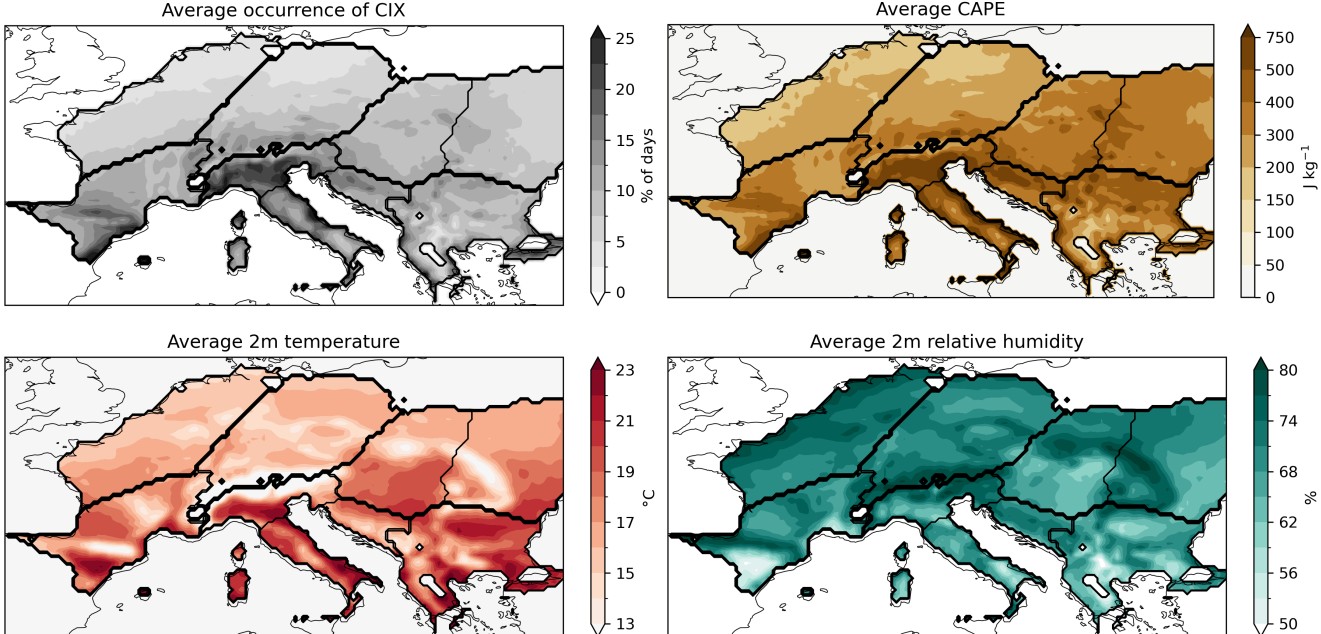

**Figure 2.** Climatological mean convective activity, CAPE, 2m temperature, and 2m relative humidity

less relative humidity. They have moderate amounts of convective activity and moderately warm temperatures (see Fig. 2, top
row). While the Western Mediterranean adjoins the Mediterranean Sea, it lies in an area still strongly affected by the Atlantic
storm tracks and is upstream of the Mediterranean. Under predominantly westerly conditions, it lies downstream of the Atlantic
and the Iberian Peninsula.

As an example of the Mediterranean region, we take a closer look at Alta-Italia, which is Europe's most active convective
region. We compare Alta-Italia to Benelux, which is the least active region. Alta-Italia has the highest climatological average
temperature and instability, approximately 5°C warmer and 300 J kg$^{-1}$ less stable than Benelux. At the same time, relative
humidity is comparable at ∼70%. Given the much higher baseline, much less temperature and moisture advection, as well
as forced lifting, are necessary to produce conditions suitable for widespread convection. In fact, the average climatological
instability throughout the entire Po Valley exceeds the threshold of 500 J kg$^{-1}$ CAPE. Point-wise, the CIX is met in Alta-Italia
approximately 25% of the time, while in Benelux, this ranges from 5-10%.

## 3.2 Large-scale descriptors of convection

To get an overview of the large-scale flow, we first inspect composites of convective precipitation, 500 hPa geopotential height,
2m temperature, and sea surface temperature (SST) on the first day of all SCO (see Fig. 3). Convective precipitation confirms
the effectiveness of the regional classification and the SCO definition with the CIX. In each region, we find a maximum in





convective precipitation values on the day of an SCO (Fig. 3, top two rows). The geopotential height is an effective descriptor of the synoptic weather pattern, allowing a comparison with e.g. Doswell and Bosart (2001). 2m temperature and SST strongly relate to instability in terms of surface heating, while SST also plays an important role as a possible moisture source.

    The geopotential height anomaly reveals synoptic, dynamical differences between the mid-latitude and Mediterranean regions (Fig. 3, second two rows). Events in mid-latitude regions are associated with a strong anomaly in geopotential height,

with a deep ridge situated slightly downstream of convection and a pronounced trough approaching (Fig. 3 regions BE, CE, and SL). This dynamical driver is stronger the further West the region is located. To a lesser degree, this is also evident in the Mediterranean regions, especially the Western Mediterranean. The further East the region lies, the less pronounced the geopotential height anomalies are, indicating that convection is related more to local rather than synoptic conditions (Fig. 3 regions EA, HE). Investigating the relative variability shows that the variability is elevated in all regions where the ridge is

located, while it is slightly reduced in the location of the trough (see Fig. A1). The relative variability is smallest for Alta-Italia and the Adriatic, indicating that they are consistently associated with much weaker geopotential height anomalies than other regions.

    We next inspect temperature anomalies (Fig. 3, third two rows). The patterns in temperature anomalies further emphasise the divide between the northern and Mediterranean regions. SCO in Benelux and Central Europe, the regions with the strongest

geopotential anomaly, are also associated with the most substantial temperature anomalies under the ridge, reaching up to 3.8 °C. Likewise, the troughs in the upper-level wave pattern are accompanied by negative temperature anomalies, albeit less pronounced than the positive anomalies. These temperature anomalies are mirrored in SST (Fig. 3, bottom two rows), indicating that the anomalies have a certain persistence (see also Section 3.3). Based on the 800 hPa wind field (Fig. 4, wind field superposed on 800 hPa relative humidity), all regions lie downstream at the 800 hPa level of positive SST anomalies. SCO

in Benelux and central Europe are associated with substantial warming of the Baltic and North Seas and, to a lesser extent, the Mediterranean. For SCO in the Slavic region, this anomaly shifts further east, with greater anomalies over the Baltic and Black Seas, as well as the Adriatic Sea. For SCO in Eastern Europe, the Mediterranean SST anomaly is weaker, and positive anomalies are primarily located over the Baltic and Black Seas, with a weak (insignificant) negative anomaly over the Western Mediterranean. The Mediterranean regions show much weaker anomalies but still generally above-average SST. SCO in

the Western Mediterranean are associated with the largest SST anomaly located just off its coast. The consistently positive land-surface and sea-surface temperatures highlight the positive correlation between temperature and CAPE (Emanuel, 2023). The warm seas emphasise the persistence of the positive temperature anomalies, as SST anomalies changes are relatively slow and require greater forcing. Additionally, the warm seas can act as a moisture source, with evaporation increasing at higher temperatures and correlating with convective activity (Morgenstern et al., 2023; Wapler and James, 2014).

    As convection is driven by moisture, we next investigate 0-7cm soil moisture, 2m relative humidity, 800 hPa relative humidity, and 800 hPa specific humidity (see Fig. 4), providing information on evapotranspiration, and atmospheric moisture profiles relevant to convective inhibition and the elevated mixed layer. Low-level moisture is key to obtaining the necessary instability and precipitable water for deep moist convection. In contrast, the elevated mixed layer (Morris, 1986; Doswell and Bosart,





**Figure 3.** Synoptic anomalies on first day of SCO. The number of SCO per region is depicted in Fig. 1b. Hatching is applied to non-significant areas. The convective region is outlined in black and labelled in the top-right of each panel.

2001) is characterised as a relatively dry and warm air layer above the boundary layer that increases convective inhibition, allowing greater instability to build. The elevated mixed layer has low relative humidity, but as it is also very warm, it does not

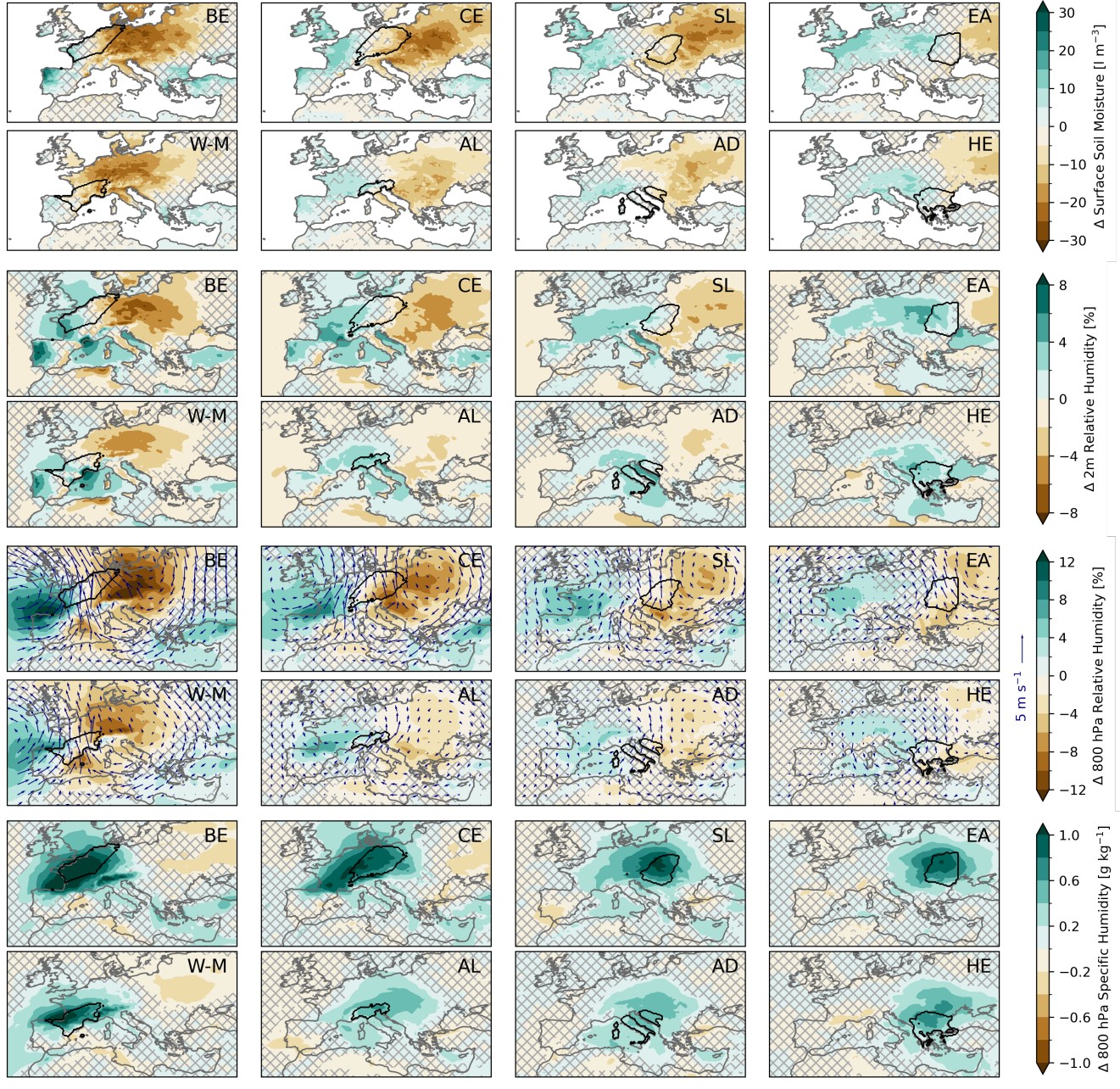

**Figure 4.** Moisture anomalies on first day of SCO. The number of SCO per region is depicted in Fig. 1b. Hatching is applied to non-significant areas. The convective region is outlined in black and labelled in the top-right of each panel. Arrows over 800 hPa relative humidity indicate wind at 800 hPa.



necessitate low specific humidity. In contrast, if specific humidity is low, deep convection may be completely inhibited.

At SCO onset, a strong soil moisture gradient is present in all regions, with a significant dry anomaly towards the East of each region, where the ridge is, and a moist anomaly approaching from the West with the beginning precipitation (Fig. 4, top two rows). The soil moisture deficits are co-located with the positive temperature anomalies (Fig. 3, third two rows). Larger temperature anomalies correspond to stronger soil dryness, indicating persistent warm and dry conditions downstream of convective outbreaks in large-scale dynamically driven regions, such as Benelux and Central Europe (for temporal evolution see Figs. 5 and 6). The soil moisture deficit also indicates a certain persistence of dry conditions downstream, as a deficit, even in the top soil layer, needs time to develop. Soil moisture gradients have been shown to play a role in convective initiation, as the moisture and temperature gradients favour a sea-breeze-like circulation, enhancing lift over the dry area (Froidevaux et al., 2014; Guillod et al., 2015; Liu et al., 2022) and the gradient (Barton et al., 2025).

2m relative humidity shows similar patterns to surface soil moisture (Fig. 4, second two rows). In comparison to soil moisture, 2m relative humidity has larger positive anomalies in the area of convection. The magnitude of both positive and negative moisture perturbations is strongest for the western and northern regions (BE, CE, W-M), moving towards weaker, but more widespread positive anomalies in the southeastern regions (EA, HE). The southeastern regions (Fig. 4, EA and HE) have the lowest 2m relative humidity in their climatology and are also associated with the largest 2m relative humidity perturbations during SCO.

The anomaly of the 900 hPa relative humidity, however, shows more pronounced dryness over the convective areas, indicating the presence of an elevated mixed layer (Fig. 4, third two rows). The anomalies are strongest in Benelux, Central Europe, and the Western Mediterranean. For these regions, it is known that elevated dry, warm air from North Africa or the Iberian Peninsula plays a role in allowing greater build-up of CAPE (Morris, 1986; Schultz et al., 2025). The composites also show exceptionally dry air over the Mediterranean, likely originating from the North African coast, as indicated by the wind field. Moreover, for Benelux and Central Europe, the wind field indicates dry airmasses may also come from regions further East, under the influence of the hot, dry ridge. Similar source regions of elevated mixed layers have also been found for the British Isles (Lewis and Gray, 2010). These anomalies weaken towards the southerly and easterly regions, skewing towards less pronounced dry anomalies, in line with the findings of the 2m relative humidity.

The specific humidity at 800 hPa is elevated for all regions, most notably for Benelux and Central Europe (Fig. 4, bottom two rows). Contrasting the relative humidity pattern, this indicates that the dryness is primarily owed to the temperature anomaly and not an absolute lack of moisture, which still permits the formation of deep moist convection. Looking at the airstreams indicated by the wind field from Northern Africa, they begin as dry anomalies, moisten after crossing the Mediterranean, and finally contribute to positive moisture anomalies (Fig. 4, BE and W-M). The southerly winds and the change from dry anomalies over the Mediterranean to moist anomalies over the continent suggest that moisture is picked up over the anomalously warm Mediterranean Sea. The relatively dry and hot but absolutely moist layer serves the purpose of creating convective inhibition and delaying the onset of convection, allowing for greater instability to build through surface heating while still allowing deep convection to take place, as sufficient moisture is available.



**Figure 5.** Hovmöller diagram of anomalies in convective precipitation, geopotential height at 500 hPa, 2m temperature, and 2m relative humidity for transient events. The number of transient SCO per region is depicted in Fig. 1b. Time evolution from 5 days pre-SCO onset to 5 days post-SCO onset is portrayed. Dashed line indicates SCO onset. Solid vertical lines indicate lateral region boundaries, regions are labelled in top-right of each panel.





To briefly summarise, we thus far find three sub-categories of regions: **Temperature-limited regions**, such as Central Europe and Benelux, where a trough-ridge-couplet induces winds advecting warm and moist air, resulting in instability. **Humidity-limited regions** such as Eastern Europe or the Hellenic Peninsula are more dependent on a positive near-surface moisture
anomaly than a strong synoptic perturbation. Lastly, **convectively favourable regions**, such as Alta-Italia and the Adriatic, show relatively small anomalies and low variability across events since their climatology already reaches favourable SCO conditions without additional perturbation, but generally slightly moister conditions than on average.

### 3.3 Transient and persistent outbreaks

Transient outbreaks are defined as single-day SCO with a gap of at least 3 days to the next SCO in the same target region.
Conversely, persistent SCO are defined as at least two consecutive days of SCO. We investigate the temporal progression of the SCO with Hovmöller diagrams of the 5 days leading up to an SCO and the 5 days after the SCO started. A meridional average is computed across the latitudes corresponding to each region. The zonal limits of each region are indicated in the diagram.

Transient SCO (see Fig. 5) show a clear and regular zonal migration of convective precipitation, the related trough and ridge, the surface temperature anomaly, and the 2m relative humidity anomaly. Convective precipitation indicates that, in many cases,
an SCO is not confined to one region but migrates from West to East over time, affecting several regions successively (Fig. 5, top two rows).

Temperature-limited regions are associated with particularly strong perturbations in geopotential height emerging just before the SCO, accompanied by positive temperature anomalies and a strong surface moisture gradient (Fig. 5, BE and CE). The presence of the ridge is longer-lasting and more pronounced than the trough. The persistence of the ridge emphasises the
230 importance of the warm anomaly and the potential presence of a dry layer for widespread convection. The surface moistening after the SCO indicates the onset of precipitation and the arrival of the trough.

Contrastingly, the convectively favourable regions show a suppression of convection on the days following the SCO (Fig. 5, AL and to a lesser extent AD). A trough resides over the region, leading to a negative temperature anomaly and a suppression of convection. In these regions, transient SCO are also the minority fraction, with most SCO lasting multiple days. The humidity-
235 limited regions show a weak moist anomaly, with no significant dry anomalies and hence no remarkable gradient (Fig. 5, EA and HE).

Persistent SCO are characterised by more pronounced anomalies in all descriptor variables (see Fig. 6). For temperature-limited regions, persistent SCO are associated with large-scale features suggesting blocking, such as a spatial stalling of the pattern in the geopotential height anomaly after SCO onset (Fig. 6, BE and CE). This stalling can also be found in temperature,
soil moisture (not shown), surface moisture, and convective precipitation anomaly as well. Especially for Benelux, after the onset of convection the pronounced, positive geopotential height anomaly remains stationary and is accompanied by persistent warm and dry anomalies. The presence of a wave train combined with blocking has also been shown for hail clusters north of the Alps (Barras et al., 2021), contrasting single-day events with smaller perturbations and no blocking anomalies. Convective precipitation persists for 2-3 days within the region, but only later affects downstream areas. This local persistence contrasts
with the signal in transient events, where convective precipitation is elevated downstream one day post-event (Fig. 5, BE and





**Figure 6.** Hovmöller diagram of anomalies in convective precipitation, geopotential height at 500 hPa, 2m temperature, and 2m relative humidity for persistent events. The number of persistent SCO per region is depicted in Fig. 1b. Time evolution from 5 days pre-SCO onset to 5 days post-SCO onset is portrayed. Dashed line indicates SCO onset. Solid vertical lines indicate lateral region boundaries, regions are labelled in top-right of each panel.





CE). The persistent SCO are associated with a greater local persistence in convection as well as in the downstream moisture deficit and heat anomaly. Transient outbreaks peak in temperature anomaly on the day of convection in the target region (Fig. 5, BE and CE). Persistent outbreaks show hotter peaks and have stronger and longer-lasting anomalies in downstream areas (Fig. 6, BE and CE). This significant co-occurrence of persistent convection, heat, and dryness indicates a contrast of weather

extremes, such as convectively-induced floods contrasting high temperatures and pronounced soil dryness on a continental scale (e.g., Tuel et al., 2022). The stalling of convection does not greatly affect its likelihood to travel from one region to the next, especially the Northern regions show a zonal migration signal, though the migration is delayed in comparison to the transient events.

The convectively favourable regions now show a more transient pattern in geopotential height, with a return to the mean state

of geopotential height and temperature after the first day of convection (Fig. 6, AL and AD). Given the convective favourability of the mean state (see Fig. 2a), it is the absence of explicit suppression, rather than the forcing of beneficial conditions, that produces the persistent event.

The humidity-limited regions show considerable positive moisture anomalies for persistent events that extend far upstream, contrasting with the rather weak anomaly for transient events. Persistent events also make up the majority of events here (see

Fig. 1).

### 3.4 Climatological and event-based trends

Given the absolute thresholds required for convection, we refrain from removing long-term trends from the data. However, the clear correlation of CAPE and convection with temperature indicates that the warming trend of the past 40 years has an effect on convection in Europe (see also Battaglioli et al., 2023; Wilhelm et al., 2024). Fig. 7 shows the decadal, 1980-2022, linear

trend of the convective proxy, CAPE, 2m temperature and relative humidity. While temperature has increased everywhere (see also Twardosz et al., 2021), CAPE trends vary regionally (see also Taszarek et al., 2021b). The regional heterogeneity of CAPE trends is primarily due to changing moisture availability. In areas where surface-level moisture is decreasing the most (France, Iberian Peninsula, Eastern Europe; see also Brogli et al., 2019, for a future climate analysis), CAPE is stable or decreasing. Where moisture availability either increases (Dinaric Alps) or decreases only slightly (Alta-Italia, Central Europe),

CAPE increases more robustly. Alta-Italia experiences the strongest positive trend, in agreement with studies highlighting the strongest observed convective trends in Northern Italy (Battaglioli et al., 2023; Taszarek et al., 2021a).

Additionally, we calculate trends of temperature and geopotential height, 2m temperature, and 2m relative humidity during convective outbreaks (see Fig. B1). For temperature- and humidity-limited regions, a strengthening of the upstream ridge relative to the convection is evident. There may also be a broadening of the ridge. The approaching trough either remains neutral or

275 deepens slightly. This behaviour is in agreement with the corresponding temperature trends, which are largely positive downstream of convection and neutral in the area of convection, resulting in a stronger overall temperature gradient.

Temperature-limited regions have a neutral moisture trend and almost no warming signal in the area of convection, maintaining the limitations on climatologically typical instability. Humidity-limited regions show a warming trend in addition to a pronounced drying, amplifying the humidity limitation. Climatologically, the convective index slightly decreases in Eastern





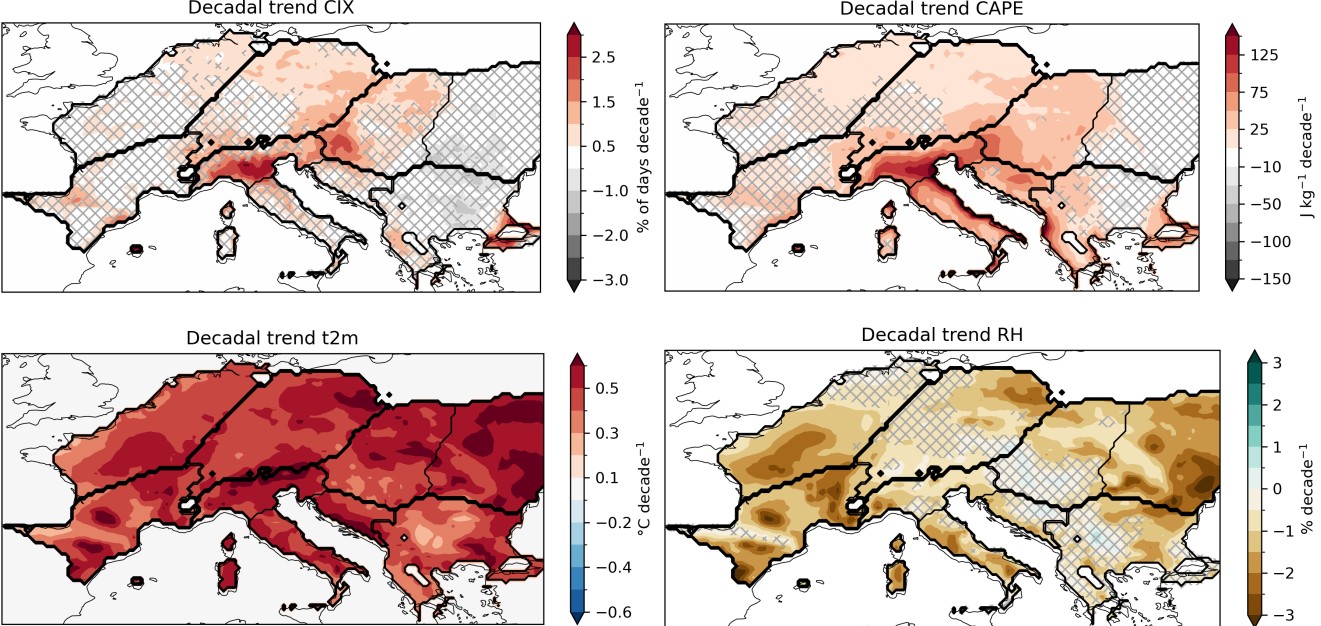

**Figure 7.** Decadal trends in CAPE, 2m temperature and 2m relative humidity, insignificant areas at p≤0.05 are hatched

Europe, indicating the less-favourable convective conditions.

Convectively favourable regions have experienced only small changes to the synoptic situation, aside from a general increase in geopotential height related to the overall warming.

Overall, the changes in convective frequency are well-aligned with changes in the climatology of temperature, moisture, and instability. We conclude that the primary driver of the trend is not owed to changes in dynamical patterns, emphasised by the fact that the dynamical patterns do not change greatly over time, aligning with the results of Ghasemifard et al. (2024), who show European hail trends are largely independent of large-scale weather trends.

## 4 Summary and Discussion

In summary, our analyses show three different types of large-scale drivers for regional SCO (severe convective outbreaks): temperature-limited types, humidity-limited types, and convectively favourable types. The most archetypal region for each class are the Benelux area for temperature-limitation, Eastern Europe for humidity-limitation and Northern Italy for convectively favourable. Central Europe falls into the temperature limited group, while the Slavic area has both hallmarks of humidity- and temperature-limitation. The Hellenic Peninsula mostly shows characteristics of humidity-limitation, albeit not as strongly as Eastern Europe, which is further removed from maritime moisture sources. The Adriatic is quite favourable for convection, though not quite as strongly as Alta-Italia. Finally, the Western Mediterranean exhibits characteristics of temperature-limitation,





while being generally more favourable for convection.

**Temperature-limited SCO** are characterized by strong perturbations in the mid-level geopotential height, lying on the western flank a pronounced ridge and at the leading edge of an approaching trough. At the surface, they are accompanied by strong positive temperature anomalies, extending from the area of SCO into the downstream ridge. Driven by the large-scale weather situation, strong shallow soil moisture gradients are present. These gradients have been shown to locally increase shear, due to

300 amplified temperature gradients causing convergence in the area (Barton et al., 2025). While soil moisture is not the primary driver of shear here, the co-occurrence of strong gradients in soil moisture, 2m temperature, and strong mid-level flow indicates an amplification of convection-favouring factors. The presence of a hot and dry area upstream of low-level flow (generally to the east of the SCO) is also relevant for the formation of an elevated mixed layer (Schultz et al., 2025). While transient (50%) and persistent (50%) events are associated with the same synoptic patterns, the synoptic patterns are more pronounced for

persistent events, and the propagation speed of patterns stalls around the onset of convection, indicating stationarity.

**Humidity-limited SCO** are associated with similar characteristics as temperature-limited SCO, albeit with weaker amplitudes in the 2m temperature, geopotential height, and soil moisture anomalies. The anomaly in 800 hPa specific humidity, however, is of the same magnitude as in the temperature-limited regions, stressing the importance of sufficient moisture in a climatologically rather dry area. Moreover, 2m relative humidity is consistently positive in the area of convection, with transitions

towards low relative humidity anomalies lying further eastward. The majority ($\sim 60\%$) of all events are persistent events. The 2m relative humidity anomaly is stronger for persistent events and extends further upstream in the days leading up to an SCO, and persisting in the region days after SCO onset. Transient events show a smaller positive 2m relative humidity anomaly and no persistence.

**Convectively favourable SCO** overall have weak synoptic anomalies, as the regions they occur in climatologically have very

favourable conditions for convection. Conditions tend to be warmer and moister than normal, but at smaller magnitudes than in the other regions. Transient SCO also constitute the minority here ($\sim 30\%$), they are associated with a cold anomaly following the SCO, preventing further widespread convection. Persistent SCO, however, show no particular sign of persisting anomalous conditions, rather a return to the already favourable mean state.

Figure 8 summarizes the identified features that are at play in all regions, but with differing magnitude.

Especially the processes relevant in temperature-limited regions show great similarity to the framework presented in Doswell and Bosart (2001), with severe convection being most likely in the prefrontal zone, in between an upper-level upstream trough and a downstream ridge. We highlight the importance of changes to the instability profile in these situations, where Doswell and Bosart (2001) focus on the amplification of deep-layer and low-level shear for tornado formation. These conceptual similarities highlight that while we focus on regional differences in Europe, the identified processes are applicable beyond this regional

limitation, and similar synoptic situations can be expected to enable SCO in appropriate regions globally.

Trend analyses show that the identified large-scale drivers do not change fundamentally in the 40-year period analysed here. While the ridge and warm anomaly downstream of an SCO tend to increase in temperature and area, the overall synoptic situation remains similar. The underlying warming and regional drying trends are consistent with trends in CAPE and frequency





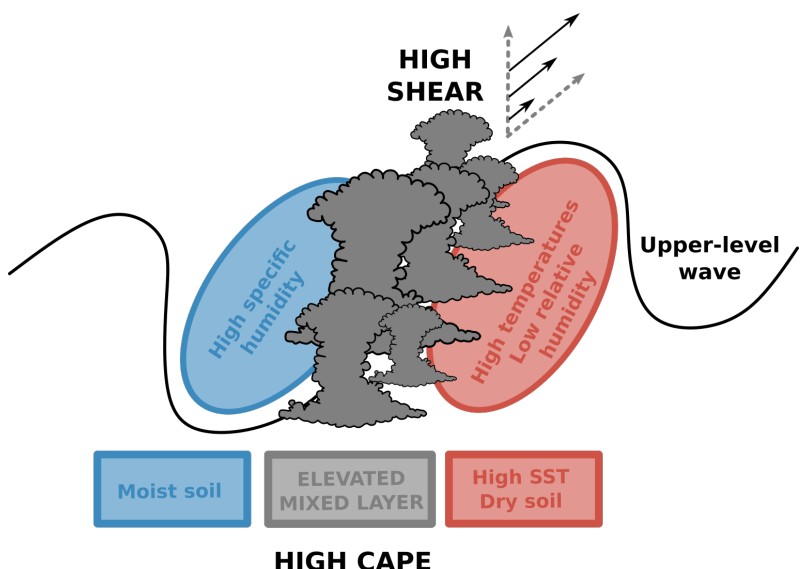

**Figure 8.** Schematic of large-scale drivers of convection over Europe

trends in CIX (convective index). As proposed by Ghasemifard et al. (2024), this indicates that convective trends are governed

by the background climate, rather than changes in the synoptic drivers.

## 5 Conclusions

This study provides a European-wide perspective on the synoptic conditions associated with SCO, identifying three main regional patterns: temperature-limited (e.g., Benelux and Central Europe), moisture-limited (e.g., Eastern Europe and the Hellenic Peninsula), and convectively favourable regions (e.g., Alta-Italia and the Adriatic).

In temperature-limited regions, strong synoptic disturbances and elevated mixed layers support the build-up of instability. In contrast, in moisture-limited regions weaker synoptic perturbations in tandem with anomalously moist low-level conditions are important. Convectively favourable regions experience more frequent SCO under near-climatological conditions with slight positive temperature and moisture anomalies. Across all regions, SCO often occur in the transition zone between a trough and a ridge, where positive temperature and moisture anomalies and moderate shear support convective organization.

This analysis of the prevalent synoptic situation during SCO in different regions of Europe provides much-needed context to better understand the regional processes leading to SCO. While previous studies have assessed regions individually, this comprehensive analysis of all regions within the same dataset allows for a quantitative comparison of regional SCO occurrence and the involved synoptic processes. The frequency of SCO differs considerably between the regions, owed to their differing

climate regimes. The climatological conditions require different perturbations to reach sufficient temperature and humidity levels to obtain instability. Despite the regional differences, the qualitatively similar patterns, as well as similarities to conceptual



frameworks from the USA (see Section 4), indicate that the fundamental processes driving SCO are applicable beyond the regions analysed here.

This type of general synoptic framework for convective-scale weather events offers new opportunities in the applications of predictability and the understanding of climate processes. In particular, predictability is strongly related to the spatial scale of weather phenomena, with synoptic-scale patterns having much greater predictability than local convective precipitation. Leveraging systematic synoptic signatures as indicators of SCO offers promising avenues for future research dedicated to improving forecasts and warnings for severe convection. Likewise, synoptic-scale processes are resolved and represented in climate simulations, especially global ensemble simulations, which currently do not support km-scale, convection-permitting resolutions. Establishing a reliable synoptic proxy can be an alternative to current analyses, which are typically based on CAPE-shear proxies and heavily depend on the available spatio-temporal resolution. While more work is required to fully establish such a synoptic proxy and test its robustness in different applications, the scientific outlook is promising on temporal scales of both weather forecasting and climate change analysis.

*Code availability.* The code is accessible via the github repository https://github.com/feldmann-m/EU_conv.



## Appendix A: Variability

To obtain the relative variability of geopotential height, we compute the standard deviation of geopotential height over the entire time period, as well as only during convective events. We then normalise the event variability with the overall variability and express the deviations in %. Positive anomalies experience more variability than normal, whereas negative areas have less variability than normal.

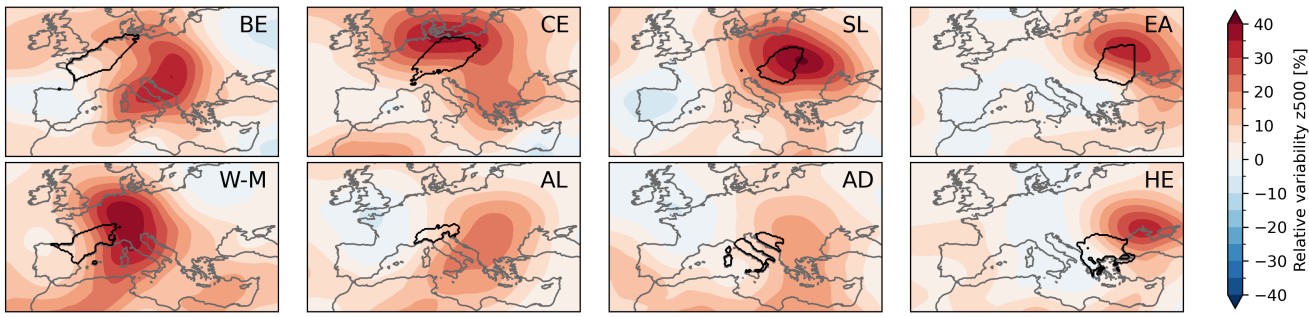

**Figure A1.** Relative variability of z500 anomaly in each region

## Appendix B: Trends

To complement the climatological trends, we apply a linear trend fit during convective events. We here provide geopotential height, 2m temperature and 2m relative humidity, to provide the variables necessary to investigate instability and changes in large-scale dynamics.

*Author contributions.* M.F. and O.M. designed the study. M.F. conducted the analyses, produced the visualizations and wrote the manuscript. O.M. and D.D. advised M.F. during the project. All coauthors revised and edited the manuscript.

*Competing interests.* M.F. and O.M. are in positions financed by the Mobiliar Insurance Group. This played no role in any aspect of this study. D.D. is a member of the editorial board of Weather and Climate Dynamics.

*Acknowledgements.* We thank Shira Raveh-Rubin, Leehi Magaritz-Ronen, Edgar Dolores-Tesillos, and Duncan Pappert for the insightful discussions on large-scale dynamics of cyclones and blockings. This project has received funding from the European Research Council (ERC) under the European Union's Horizon 2020 research and innovation programme (grant agreement No. 847456). We acknowledge the use of ChatGPT for code debugging and minor text revisions in the manuscript.



**Figure B1.** Trends of geopotential height, 2m temperature and 2m relative humidity during convective outbreaks, hatching indicates non-significant areas

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
