# Peer review of "A pan-European analysis of large-scale drivers of severe convective outbreaks"

_EGUsphere, 2025_

## Referee Comment (RC2)

**Review of: A pan-European analysis of large-scale drivers of severe convective outbreaks**

This study investigates the large-scale drivers of severe convective outbreaks (SCO) in Europe. SCO events cause significant damage to infrastructure, property, and human health and life, and they occur regularly across Europe. However, while a well-defined model of severe convection exists for the US, other extratropical regions like Europe still lack a comprehensive framework. This paper therefore represents an important step toward improving SCO predictability, and even understanding how these events may evolve under global warming.

The authors detect convective events using a CAPE-shear threshold binary variable (CIX) and cluster them into regions where SCOs occur simultaneously. These regions are then grouped into three categories based on the dominant climatological perturbation driving SCO initiation during the extended summer. Using this framework, the paper examines the large-scale descriptors of convective outbreaks, differences between short-lived and persistent SCOs, and trends in these descriptors.

The paper is well-organized, clearly written, and methodologically sound. I particularly commend the authors for their rigorous use of statistical inference, including the application of False Discovery Rate (FDR) correction for gridded testing—a theoretically necessary but often overlooked practice in climate studies.

Overall, the manuscript requires only minor revisions. I have read the comments from Reviewer #1 and agree with their suggestions, particularly regarding the inclusion of Appendix figures (especially B1) in the main text. Below are my additional remarks:

-   Line 70: The use of CIX alone is one of the relatively weaker aspects of the study. As the authors mention comparing CIX-based clustering with lightning data, I recommend including this comparison in the Appendix for transparency.

-   Line 78: The authors note that they tested different CAPE and shear thresholds for defining CIX. It would be valuable to demonstrate the robustness of the results to these threshold choices, especially given the lack of validation for CIX as a convection proxy using observations.

-   Line 138: The statement "*In each region, we find a maximum of convective precipitation on the day of the SCO*" is technically correct but initially confusing when examining Fig. 3. At first glance, one might expect the precipitation maxima in panels (first two rows) to align strictly with the detected SCO regions. Instead, the key finding is that each region experiences its peak convective precipitation during its respective SCO events. I suggest clarifying this point and briefly discussing overlaps between regions (e.g., W-M/AL SCOs sometimes coincide with convection in BE/CE or SL/AD).

Additional minor corrections:

- Line 193: 900 hPa → 800 hPa

- Line 163 & Fig. 7 caption: Replace "insignificant" with "non-significant" or "not statistically significant" for precision. "Insignificant" can be misleading, especially when used in a text with no reference to the result of a statistical test as at Line 163.

---

## Author Comment (AC1)

We thank both reviewers for their careful review of the manuscript. To address the concerns, an additional comparison to observational lightning data has been added. Additionally, tables containing quantitative regional differences are now included and discussed in the results. Moreover, all requested figures were moved from the appendix to the main manuscript. The discussion was modified to reflect the desired nuances on saturation in comparison to specific humidity. We thank the reviewers for their advice and support. Below are all point-by-point responses. Line numbers refer to the tracked-changes manuscript, which is included at the end of the document.

**A pan-European analysis of large-scale drivers of severe convective outbreaks**

Authors: Feldmann et al.

Severe convective outbreaks (SCO) are important drivers of weather-related damages in Europe. Hence, it is important to understand which large-scale drivers establish convectively favorable conditions to SCO in different regions, offering new opportunities in the applications of predictability and the understanding of climate processes since synoptic-scale patterns have much greater predictability than local convective precipitation. Although more work is required to establish a reliable synoptic proxy as an alternative to current CAPE-shear-based analysis, the scientific outlook is promising.

The paper nicely deals with this topic, identifying three macro-regions, affected in a different way by the large-scale patterns, and discussing land-surface conditions that are generally not considered in this kind of studies. The presentation is overall clear, and I like the concise and essential writing style. Thus, I think that the paper can be accepted after some relatively minor modifications, related both to improvements in the presentation and in the discussion of the results.

Major points:

1.  L212-217: What I really miss in the article is a quantification of your claims. I think adding a table (even in the Appendix) to quantify the anomalies in Figures 3 and 4 and the trends in Figure B1 of the different variables in each identified region could help support your analysis. For example (L279), you mention "a pronounced drying" in the humidity-limited regions, but I do not see a more pronounced drying than in the temperature-limited regions: can you quantify the trends in a table? This would make the summary and discussion in Section 4 more robust.
    In the revisions, the Figure numbers have been adapted. The anomaly patterns in the new figures 5, 6 and 10 (previous 3, 4 and B1) are quite large also outside of the

defined regions. E.g., the z500 anomalies are largest to both the east and west of each regional polygon, and the t2m anomaly is strongest quite far east of each region. Hence a simple magnitude of an anomaly over either the regional domain, or the whole European domain is not very meaningful, especially for gradients. Rather, it is important to look at the spatial distribution of each variable individually. All regions are shown with the same colorbars, so their panels are directly comparable. We instead opted to include the climatological average and trend for each region in additional tables (Tables 2 and 3).

Importantly, the saturation-limited regions do not necessarily have absolutely larger moisture-related anomalies, than e.g. the temperature- and dynamics-related anomalies. Given the climatologically drier and warmer climate, increasing saturation is relatively more important than increasing temperature to achieve instability. We have further clarified this point in the text throughout the manuscript.

2.  L272: I think that here and in most of the paper there is an excessive emphasis on relative humidity rather than on specific humidity; however, RH is not a measure of humidity but rather of the proximity of the environment to saturation. So, why not adding specific humidity in Fig. 7?

    We focus the discussion on relative humidity, as saturation is a key component in achieving instability. To avoid confusion, the second macro-region is now called saturation-limited. We further adapt the phrasing in the manuscript in various locations to reflect this focus on saturation, rather than absolute moisture content. We do include here the specific humidity climatology and trend at 900 hPa in Figure R1 below. Specific humidity is increasing throughout the regions analyzed (or stable) and for the evolution of CAPE, the evolution of saturation is more important. We therefore think that the figure does not add important material to the discussion and since the paper is already quite long, we would prefer not to include this figure in the paper.

[Figure]

    Figure R1: Average 900 hPa specific humidity in Europe (left panel), and decadal trend of 900 hPa specific humidity (right panel) from 1980-2022.

3.  I do not understand the need to put Figure B1 (and in minor way Fig. A1) in the Appendix, as it is an important part of the study.

    Both of these figures have been moved to the main manuscript (Figs. 5 and 10)

Minor points:

- L11: delete "climate"
  Removed

- L48: which is the average duration of a persistent event?
  45% of persistent events last 2 days and an additiona 35% up to 4 days, decaying with an approximately exponential distribution. This has been added to the text (line 115 ff).

- Table 1: why is the average and not the maximum 2m temperature considered?
  We performed analyses with both and obtained very similar results. We opted to show the results of average 2m temperature, as this is more consistent with the processing of the other variables. For CAPE and the CIX/SCO definition it is necessary to use the daily maximum, as CAPE behaves very nonlinearly on a diurnal basis.

- Figure 1: I do not think the selected names are representative of the regions considered: what about BA (Balkanic peninsula) or E-M (eastern Mediterranean) instead of HE, C-M (central Mediterranean) instead of AD, N-W (north-western region) instead of BE?
  The regions have been renamed throughout the text, as well as the abbreviations in the figures and tables.

- Figure 1 caption: panels b) and c) are reversed between the figure and the caption
  This has been corrected.

- L86: I think that "To calculate the anomalies, …" should be put before "All descriptor variables"
  This has been adapted accordingly (line 88).

- L106: Also, squall lines frequently cross the Po valley from west to east (e.g., De Martin et al. (2024))
  De Martin F., Davolio S., Miglietta M. M., and Levizzani V., A conceptual model for the development of tornadoes in the complex orography of the Po valley, Mon. Wea. Rev., 152, 2024, 1357-1377, https://doi.org/10.1175/MWR-D-23-0222.1
  This has been added (line 109f).

- L108: does a high value of SCO necessarily mean that an outbreak occurs? Please shortly discuss this point
  As reviewer 2 requested, we now include a composite of anomalous lightning activity for the time period, where lightning data is available (see Fig. R2, and manuscript Fig. 3). We added a short discussion of this qualitative validation in the

manuscript (line 146ff).

[Figure]

Fig. R2: Anomalous lightning occurrence during first day of SCO per region.

- L119: "when absolute temperatures are high" is redundant
  This sentence has been rephrased (line 127).
- L124: "left column" instead of "top row"
  This has been changed accordingly (line 133).
- L132: "climate average" instead of "baseline"
  This has been rephrased (line 141).
- L132: please add "anomalies" after "much less temperature"
  This sentence has been rephrased (line 141).
- L133: forced lifting depends also on CIN, which is not considered here
- L149-152: how do you explain the high variability in HE?
  The regions with particularly low variability are the convectively favourable ones, not the saturation- or temperature-limited ones. We now clarify this in the text. The overall variability of geopotential decreases towards the south and east in Europe. With a similar geopotential anomaly as AL/AD, HE can still achieve higher relative variability.
- L157: really, only the positive temperature anomalies are mirrored in SST
  This has been adapted accordingly (line 177).
- L159: the sentence is ambiguous: in my opinion, all regions lie downstream of positive SST anomalies but only of the Mediterranean Sea. This makes some confusion with the following sentences.
  This has been further clarified to "Mediterranean and Black Seas", as e.g. Eastern Europe is downstream of the Black Sea and affected by its positive SST anomaly (line 180).
- L166: add "anomalies" after sea-surface temperatures
  This has been added (line 187).
- L206-208: although not necessary, it may be interesting to investigate the sea surface fluxes to support this hypothesis.

> The Era5 dataset we currently have downloaded does not contain sea surface fluxes. In the interest of time, we refrain from this additional analysis.

- L237-249: all this part can be significantly reduced; there are some points repeated a few times.

  > This section has been shortened.

- L262: "Given the absolute thresholds required for convection, we refrain from removing long-term trends from the data.": it is not clear to me what you mean here

  > This motivates, why the data was not detrended in the first place. We rephrased the sentence accordingly (line 286).

- L269: "or decreases only slightly (Alta-Italia, Central Europe)": I would say it occurs in Slavic area rather than in Central Europe

  > Both the Slavic and Adria areas were now included (line 294f).

- L273: "upstream": or downstream? Or both upstream and downstream?

  > This has been changed to "to the east of convection" to avoid confusion (line 301).

- L284-286: I think this is an important conclusion of the paper and deserved a bolded font

  > While we refrain from putting the whole sentence bold here, we add this in the conclusion, to highlight it as an important point (line 368).

- L314: remove "have"

  > This sentence has been rephrased (line 342f).

**Review of: A pan-European analysis of large-scale drivers of severe convective outbreaks**

This study investigates the large-scale drivers of severe convective outbreaks (SCO) in Europe. SCO events cause significant damage to infrastructure, property, and human health and life, and they occur regularly across Europe. However, while a well-defined model of severe convection exists for the US, other extratropical regions like Europe still lack a comprehensive framework. This paper therefore represents an important step toward improving SCO predictability, and even understanding how these events may evolve under global warming.

The authors detect convective events using a CAPE-shear threshold binary variable (CIX) and cluster them into regions where SCOs occur simultaneously. These regions are then grouped into three categories based on the dominant climatological perturbation driving SCO initiation during the extended summer. Using this framework, the paper examines the large-scale descriptors of convective outbreaks, differences between short-lived and persistent SCOs, and trends in these descriptors.

The paper is well-organized, clearly written, and methodologically sound. I particularly commend the authors for their rigorous use of statistical inference, including the application of False Discovery Rate (FDR) correction for gridded testing—a theoretically necessary but often overlooked practice in climate studies.

Overall, the manuscript requires only minor revisions. I have read the comments from Reviewer #1 and agree with their suggestions, particularly regarding the inclusion of Appendix figures (especially B1) in the main text. Below are my additional remarks:

- Line 70: The use of CIX alone is one of the relatively weaker aspects of the study. As the authors mention comparing CIX-based clustering with lightning data, I recommend including this comparison in the Appendix for transparency.
  We now include an explicit composite of lightning-occurrence anomaly to highlight the appropriateness of our event definition (see Fig. R3 below and manuscript Fig. 3, line 146ff). The designated convective regions all show significant positive anomalies, stressing that there is in fact observed convective activity. The lightning data only covers half of the analysis period, so we refrain from pulling it through further analyses, as it is not directly comparable to the ERA5 data.

[Figure]

Fig. R3: Anomaly of lightning activity on first day of SCO per region.

Lightning anomalies fit better to the target regions than convective precipitation. Convective precipitation is also active in frontal regions, which generally lie to the west of the SCO – and in the convective precipitation composite, positive anomalies also lie to the west of the target region. This spill-over to other regions is more related to the ambiguous nature of the convective precipitation variable, which corresponds to the convective parametrization being active, rather than representing the desired SCO activity directly.

- Line 78: The authors note that they tested different CAPE and shear thresholds for defining CIX. It would be valuable to demonstrate the robustness of the results to these threshold choices, especially given the lack of validation for CIX as a convection proxy using observations.
  Since we now provide an additional comparison to lightning data, we refrain from further elaborating on the sensitivity of the chosen thresholds in the manuscript.

- Line 138: The statement "*In each region, we find a maximum of convective precipitation on the day of the SCO*" is technically correct but initially confusing when examining Fig. 3. At first glance, one might expect the precipitation maxima in panels (first two rows) to align strictly with the detected SCO regions. Instead, the key finding is that each region experiences its peak convective precipitation during its respective SCO events. I suggest clarifying this point and briefly discussing overlaps between regions (e.g., W-M/AL SCOs sometimes coincide with convection in BE/CE or SL/AD).
  Yes, the approach does not prohibit two regions from having an SCO on the same day. However, the lightning data shows that this is more owed to the ambiguity of convective precipitation, rather than actual convective activity. We include an additional clarifying paragraph in the manuscript (line 154ff).

Additional minor corrections:

- Line 193: 900 hPa → 800 hPa

- Line 163 & Fig. 7 caption: Replace "insignificant" with "non-significant" or "not statistically significant" for precision. "Insignificant" can be misleading, especially when used in a text with no reference to the result of a statistical test as at Line 163.

[revised manuscript text omitted]